# Effects of Trail Running versus Road Running—Effects on Neuromuscular and Endurance Performance—A Two Arm Randomized Controlled Study

**DOI:** 10.3390/ijerph20054501

**Published:** 2023-03-03

**Authors:** Scott Nolan Drum, Ludwig Rappelt, Steffen Held, Lars Donath

**Affiliations:** 1Department of Health Sciences—Fitness Wellness, College of Health and Human Services, Northern Arizona University, Flagstaff, AZ 86001, USA; 2Department of Intervention Research in Exercise Training, German Sport University Cologne, 50933 Cologne, Germany

**Keywords:** postural balance, gait, agility, muscle strength, long distance running, endurance training, running surface

## Abstract

Running on less predictable terrain has the potential to increase the stimulation of the neuromuscular system and can boost aerobic performance. Hence, the purpose of this study was to analyze the effects of trail versus road running on neuromuscular and endurance performance parameters in running novices. Twenty sedentary participants were randomly assigned to either a trail (TRAIL; n = 10) or road running (ROAD; n = 10) group. A supervised and progressive, moderate intensity, and work-load-matched 8 wk endurance running program on TRAIL or ROAD was prescribed (i.e., randomized). Static balance (BESS test), dynamic balance (Y-balance test), gait analysis (RehaGait test, with regard to stride time single task, stride length dual task, velocity single task), agility performance (*t*-test), isokinetic leg strength (BIODEX), and predicted VO_2_max were assessed in pre- and post-tests. rANOVA analysis revealed no significant time–group interactions. Large effect sizes (Cohen’s d) for pairwise comparison were found for TRAIL in the BESS test (d = 1.2) and predicted (pred) VO_2_max (d = 0.95). Moderate effects were evident for ROAD in BESS (d = 0.5), stride time single task (d = 0.52), and VO_2_max predicted (d = 0.53). Possible moderate to large effect sizes for stride length dual task (72%), velocity single task (64%), BESS test (60%), and the Y-balance test left stance (51%) in favor of TRAIL occurred. Collectively, the results suggested slightly more beneficial tendencies in favor of TRAIL. Additional research is needed to clearly elucidate differences between TRAIL and ROAD, not only in novices but also in experienced exercisers.

## 1. Introduction

Regular physical activity, such as running, enhances cardiorespiratory and neuromuscular performance and is associated with a delay in all causes of mortality and morbidity [1,2,3,4]. Lee et al. [5] found that minimal running training volumes of 30–59 min a week, or 5–10 min a day are associated with lower risks of all-cause and cardiovascular mortality. Despite proven health benefits of physical exercise, the number of sedentary people worldwide is large and steadily growing [6,7,8] in both sexes and with increasing age [7,9]. Physical inactivity accelerates aging-induced functional decrements and compromises physical performance which can lead to impairments in activities of daily living [3,10,11]. 

At approximately 30 years of age, muscle mass and muscle strength begin to decrease gradually by 10–15% each decade [3]. Progressive skeletal muscle atrophy is accompanied by a loss in muscle coordination and a decline in balance [11], which can already be evident in individuals of ≥40 years of age [12]. Balance impairments and related spatiotemporal gait deficits both represent crucial risk factors for falls and fall-related injuries [13,14,15]. Falls and fall-related injuries as well as general health impairments not only occur in the elderly but are a frequent problem in middle-aged and young people [16,17]. Few studies have investigated falls and the frequencies of falls in young and middle-aged individuals [16]. In a longitudinal study by Niino et al. [17], the prevalence of falls among middle-aged individuals (40–59 years) was 12.9%, compared to 16.5% among the elderly group (60–79 years). Talbot et al. [16] observed a prevalence of one or more reported falls within a two-year period in 18.5% of young adults, 21% of middle-aged adults and 35% of older adults. In addition to the direct consequences of falls, many people develop a fear of falling after such an event which often leads to a vicious cycle of reduced physical activity, decreased mobility and muscle strength, and a subsequent higher risk for future falls [14,18,19].

To refute the natural decline in neuromuscular properties with aging and augment spontaneous balance and maintenance of strength, our main study objective was to determine the effectiveness of exercising on uneven surfaces (i.e., dirt trails) vs. familiar (or predictably even road) surfaces in a younger adult population on the prior mentioned variables (e.g., neuromuscular or gait training, balance, strength). For instance, running has been shown to improve or amplify several task-specific, metabolic, and neuromuscular factors [20]. However, few studies have focused on neuromuscular variables (e.g., gait parameters via a wearable gait analysis system) resulting from endurance training on distinctly different surfaces [20]. As a suggestion, future researchers should theoretically look at the protective effects of frequent running on uneven surfaces related to unexpected falls, especially in the elderly. Ultimately, the impact of trail running, which is attracting an increasing number of recreational and competitive runners [21,22], compared to road running, has not been extensively compared.

In the present project, we hypothesized trail running would lead to more pronounced improvements in neuromuscular and endurance performance than road running. These assumptions are based on the different characteristics of surface type and gradients between the two conditions. Trail running tends to invoke higher challenges for the neuromuscular system, especially regarding involved muscle coordination, proprioception, and activation [23,24,25,26] compared to road running. Furthermore, since uphill running is an effective stimulus for improving endurance running performance [27,28] and submaximal running economy [27,29] we expected a more pronounced performance at posttest in the submaximal incremental treadmill test for TRAIL.

## 2. Materials and Methods

### 2.1. Participants and Experimental Setting

This pilot study adheres to CONSORT guidelines [30]. Participants were recruited via flyers, posters, word-of-mouth, and local advertisement as well as via “batch” emails among faculty and staff at the university where the project was conducted. Inclusion criteria [31] for participation were: (i) 18–59 years of age; (ii) currently sedentary or not exercising more than twice a week for the last three months; (iii) free from any injury or illness and currently no intake of any medication; (iv) and non-smoker. Importantly, according to ACSM, sedentary, healthy (e.g., free of disease, non-smoker, uninjured) individuals will showcase a greater physiological change from pre to post exercise intervention. 

To ensure that participants met the inclusion criteria, all subjects were asked to complete several physical activity questionnaires. The questionnaires included: (a) International Physical Activity Questionnaire—Short Form (IPAQ-SF) [32], (b) the Physical Activity Readiness Questionnaire (PAR-Q&YOU) [33], and the (c) American College of Sports Medicine (ACSM) Risk Stratification [31] to assess individual current health and activity levels. If a participant reported two risk factors related to cardiovascular diseases, he/she had to consult a physician for medical clearance to participate in moderate to vigorous exercise.

The study was conducted according to the Code of Ethics for Human Experimentation of the World Medical Association and the Declaration of Helsinki [34]. Participants were informed in detail about the design of the study, including the potential risks and benefits of included procedures, before providing their informed written consent to participate. The study protocol was approved by the Institutional Review Board of the Northern Michigan University (Trial registration number: ID Proposal Number HS16-786; Date of registration: September, 2017).

Participants were anonymously assigned by the researcher via simple randomization using a random number generator to either TRAIL (n = 20) or ROAD (n = 19) and entered into an endurance exercise program. The program consisted of 8 weeks of gradually increasing running workouts with a total amount of 29 training sessions. This randomized controlled pilot trial compared two training groups (i.e., TRAIL vs. ROAD) in terms of balance, gait, agility, along with strength and endurance performance measures in a pre- and post-intervention testing format. Participants in the TRAIL group ran outdoors on uneven and soft trails with varying gradients and under-foot terrain (e.g., rocks, roots, more consistent undulating routes). Participants from the ROAD group ran on predictable terrain or roads with asphalt, concrete or paved surfaces exhibiting no or infrequent gradients. An adherence rate of a minimum of 80% (24 runs) was required for inclusion in the final analysis. 

To confirm, a total of 39 healthy adults were initially assigned, whereof 6 subjects did not start the program; 5 participants dropped out during the intervention due to injuries; 3 participants did not meet the required 80% adherence rate and 1 participant was not available for post testing. Additionally, 2 participants (i.e., “4” total) from each group were excluded from analysis due to other exclusion criteria—not following the prescribed training load and for participating in additional training during the period of the study. Then end total of analyzed participants equaled 20.

Demographic data at baseline for all participants who received the allocated intervention are depicted in Table 1.

### 2.2. Experimental Design

Qualifying participants were asked to report to an Exercise Science Laboratory for pre- and post-intervention testing. Post-testing sessions were scheduled at a similar time of the day as pre-testing and within a week upon completion of the training program in November and December 2017, depending on pre-testing dates. Testing order, as well as the examiner were kept constant for each participant. 

Finally, ten participants in each group were included in the statistical analysis. The study flow is depicted following the CONSORT criteria, which is easily referenced [30]. Notably, 10 participants in each group provided significant differences (alpha error probability: 0.05) and notable study power (i.e., 1-beta error probability: 0.9) when moderate to large effects size differences between group were presumed for balance performance as the primary outcome.

Lastly, mandatory running meetings were held twice a week and coaching appointments were scheduled as required. Furthermore, participants were contacted by email or phone once a week for feedback. As an additional motivation, a final joint 5k running event was held upon completion of the intervention.

### 2.3. Heart Rate and Blood Pressure

Prior to baseline testing, a blood pressure cuff (Adcuff™, Hauppauge, NY, USA) and stethoscope (Littmann, St. Paul, MN, USA) were employed for blood pressure measures; then, pre-exercise resting heart rate (Polar monitor and watch, Lake Success, NY, USA), as well as body height (Seca stadiometer, Chino, CA, USA) and weight (Health O Meter scale, Mccook, IL, USA) were measured. Maximal heart rate (HRmax) in beats per minute (bpm) was predicted using the following formula according to Tanaka et al. [35]: 207—(age × 0.7) for men and 206—(age × 0.88) for women. The lateral preference inventory for measurements of footedness [36] was used to evaluate leg dominance. Limb length was measured from the umbilicus to the medial malleolus of the right leg using a tape measure [37]. Blood pressure, pre-exercise resting heart rate, as well as body height and weight measurements were repeated before post-testing as well.

### 2.4. Warm-Up

Warm-up consisted of walking on a treadmill for 5 min at a rate of perceived exertion (RPE) of 3 on the Borg CR-10 scale [38], followed by dynamic stretching and muscle activation (Knee Hug to Forward Lunge–Elbow to Instep, Heel to Butt Moving Forward with Arm Reach, Handwalk, Lateral Squat Low).

### 2.5. Static Balance Testing

Static balance was tested with the Balance Error Scoring System (BESS) [39], which evaluates 3 stance variations in the following order: (1) double leg, (2) single leg, and (3) tandem or feet in line with one another. The test takes place on 2 different surfaces, starting on firm for all “3” conditions and ending on foam for all “3” conditions while wearing no shoes. Each trial lasts 20 s, during which the number of deviations from the proper testing position were counted. Deviations from the proper testing position in the BESS test are: (a) moving hands off the hips; (b) opening the eyes; (c) step, stumble or fall; (d) abduction or flexion of the hip beyond 30°; (e) lifting the forefoot or the heel off of the testing surface; and (f) remaining out of proper testing position for more than 5 s. Proper position consists of the hands on the iliac crest, eyes closed, and consistent foot position. For the double leg stance, feet need to touch and remain flat on the testing surface. For the single leg stance, the participant stands on the non-dominant leg with the other leg held in approximately 20° of hip flexion, 45° of knee flexion, and neutral position in the frontal plane. For the tandem stance, one foot is placed in front of the other with the heel of the anterior foot touching the toes of the posterior foot, and the non-dominant leg in the posterior position. The maximum amount of errors for any single condition was set at 10. If multiple errors were committed simultaneously, only one was recorded. To improve reliability, the test was repeated 3 times by the same examiner [39] and the mean score of the three trials was calculated for final analysis.

### 2.6. Dynamic Balance Testing

The Y Balance test (YBT) was performed to evaluate dynamic postural stability and functional symmetry during single leg stance in three (anterior, posteromedial, posterolateral) directions [40]. In a Y pattern, each posterior line was marked with tape 135° from the anterior line and 90° apart from one another. Subjects performed a practice trial followed by three test trials for each direction and each leg and were instructed to reach as far as possible, thereby pushing a pen held by the examiner to mark the reaching distance. The testing order started with standing on the left foot and reaching in the anterior direction followed by the trials standing on the right foot for the same direction. This procedure was repeated for all directions. Trials were considered invalid and were repeated if the participant either made a heavy touch or rested the reaching foot on the ground, could not return in a controlled way to the starting position, raised or moved the stance foot, or kicked the marker with the reaching foot to gain more distance [40]. Results were calculated as a composite score with the help of following formula:(((anterior length + posteromedial length + posterolateral length)/3 × leg length) × 100).(1)

### 2.7. Gait Analysis

Spatiotemporal gait parameters (stride time [s], stride length [m], and stride velocity [m/s]) were measured during 20 m (65.6 feet) of level walking at self-selected habitual walking speed by using the portable gait analysis system RehaGait^®^ (Hasomed GmbH, Magdeburg, Germany). The RehaGait^®^ system consists of two mobile sensors which are attached to the lateral part of each shoe to measure linear acceleration, angular velocity, and the magnetic field of the foot at a sampling rate of 500 Hz [41]. Each participant performed a familiarization trial followed by 2 trials with single task condition and 2 trials with dual task condition. For dual task trials, participants were asked to perform a double-digit subtraction task while walking. The combination of gait analysis with cognitive interference tasks was applied to distract participants and limit the cognitive resources for gait control. The mean score for each condition was included in further analysis. For all trials, the phases of gait initiation and deceleration at the end of the walkway were excluded from analysis. For both pre- and post-testing, participants were wearing their running shoes.

### 2.8. Agility Testing

The *t*-test evaluates the subjects’ agility, leg power and leg speed [42]. Four cones are set out in a T pattern. The test starts at the first cone with a forward sprint of 9.14 m to the second cone, continues with shuffling sideways for 4.57 m to another cone on the right, then 9.14 m to the one on the left, and again 4.57 m back to the middle, before ultimately running backwards 9.14 m to return to the starting point. The base of the cone always has to be touched with the hand further away from the cone when performing the test. The fastest out of 3 trials was used for analysis.

### 2.9. Strength Testing

Unilateral isokinetic concentric leg strength was assessed for the dominant leg using the BIODEX Multi-Joint System 4 Pro (Biodex Medical Systems, New York, NY, USA). Knee extension and flexion as well as ankle plantar- and dorsi-flexion were tested for peak torque (PT) and total work (TW). Subjects were seated with chair and dynamometer position at 90° and the dynamometer positioned outside the testing leg. The anatomical axis rotation (lateral femoral condyle on a sagittal plane for the knee and through the body of talus, fibular malleolus, and tibial malleolus for the ankle) was in alignment with the dynamometer shaft for both knee and ankle attachment, ensuring that the testing pattern was consistent with the proper biomechanics of the joint. Body parts on either side of the tested joint were firmly secured with straps, in order to restrict motion as much as possible to the area of interest. Range of motion was set for each subject and joint individually. After a 12-repetition warm-up trial at 180°/s and low effort, participants performed two sets of 5 repetitions at 60°/s and maximal effort with a 60 s break between sets.

### 2.10. Aerobic Endurance Testing

Oxygen consumption was measured by indirect calorimetry on a treadmill during the walking-based Pepper protocol [43] with the Parvo Medics TrueOne 2400 automated gas analysis system (Sandy, UT, USA). The Pepper protocol is an incremental submaximal test starting at an inclination of 0% and a velocity of 2.5 mi (4 km) per hour. Intensity increases each minute by elevating either inclination or velocity. The test is ended at 85% of predicted HRmax [35]. Gas exchange variables (VO2 and VCO2, RER), RPE on the Borg CR-10 scale [38] and HR were monitored and averaged to 15s time-intervals. Finally, maximal oxygen consumption (VO_2_max) was predicted from the highest value recorded at HR85% using the formula VO_2_max pred = VO_2_max at HR 85%/85 × 100. Prediction was used to minimize cardiovascular risk of pushing to maximum in this mixed age group (2).

### 2.11. Training Program

The training program started with 3 training sessions per week in weeks 1–3. Each training session had a duration of 22–36 min (which was the standard range throughout most of the 8 wk intervention) of running interspersed with 2 min walk rest periods. Novice participants progressed to 4 running (with prescribed intermittent walking) sessions per week in weeks 4–6 with 2-min walk breaks before gradually omitting the walk breaks in week 7 and finishing the program at the end of week 8 with a 45 min continuous run (i.e., their 4th run of week 8). Exercise training started for each participant after the pretest and was performed individually on self-selected outdoor trails (i.e., TRAIL) and roads (i.e., ROAD) at a perceived exertion of 3–4 on Borg CR-10 (although the average RPE approached “5” for both groups upon end analysis). Each participant was provided with a running log in which they recorded training duration, perceived exertion levels, location, and estimated percentage of each session on TRAIL or ROAD. Actual training loads for both groups are summarized in Table 2.

### 2.12. Statistical Analysis

Group means of all variables for all pre- and post-test data were calculated based on individual test scores in order to compare changes between groups. All data are presented as means with standard deviations (SD). Data analysis was computed using the statistical software program SPSS for Windows V.14.0 (SPSS Inc., Chicago, IL, USA). After adjustment for baseline scores (note, baseline values were added as covariate in order to adjust for potential baseline differences), repeated-measures ANOVA procedures were conducted to determine significant between-group differences. Group (TRAIL and ROAD) served as the between-subject factor, and time (pre- and post-test) as the within-subject factor. Statistical significance level was set at *p* < 0.05. Because of the small sample size, partial eta squared (ηp2) and Cohen’s d (d < 0.2 = trivial effect; d ≥ 0.2 = small effect; d ≥ 0.5 = moderate effect; d ≥ 0.8 = large effect), as the standardized mean difference, were calculated to estimate effect sizes from pre- to post-testing for all ANOVAs. The probability for an effect being practically worthwhile in favor of either TRAIL or ROAD was calculated according to the magnitude-based inference (MBI) approach (25–75%, possibly; 75–95%, likely; 95–99.5%, very likely; >99.5%, most likely) using the Hopkins [44] spreadsheet for analysis of controlled trials with adjustments for a predictor in Microsoft^®^ excel.

## 3. Results

In review, of the 33 subjects that received the allocated intervention, 5 people (4 in TRAIL; 1 in ROAD) ended the program prematurely due to injuries and/or pain. A total of 3 people (2 in TRAIL; 1 in ROAD) did not meet the required attendance rate and 1 person from ROAD never reported to the post-testing. Two more subjects of each group were excluded from further evaluation based on exclusion criteria (age, amount of previous physical activity, adherence rate, ≤2 risk factors according to the ACSM Risk Stratification). A total of 10 participants from each group were included in the final analysis. Higher baseline test scores and differences between the two groups were seen for leg strength in knee flexion PT (19.9% higher in TRAIL) and ankle plantar flexion PT (18.5% higher in TRAIL), and for VO_2_max pred (24.3% higher in ROAD).

The mean overall attendance rate for the intervention was 93.8% or 27.2 ± 2.3 out of 29 total trainings; 91.4% (26.5 ± 1.7) for TRAIL and 96.2% (27.9 ± 2.6) for ROAD.

### 3.1. Static and Dynamic Balance

The repeated-measures ANOVA revealed no statistically significant differences between groups for any balance measures. However, for the BESS test, a significant time-effect between pre-and post-testing was noted (*p* = 0.001, ηp2 = 0.46) and large and moderate effect sizes according to Cohen’s d for TRAIL (d = 1.2) and ROAD (d = 0.5), respectively. Results for static and dynamic balance testing are presented in Table 3.

### 3.2. Gait

The spatiotemporal gait analysis rANOVA showed no notable improvements over time in any parameter for either TRAIL or ROAD, as displayed in Table 4. According to Cohen’s d, a moderate effect size for stride time ST in ROAD (d = 0.52) as well as small effects for velocity DT in TRAIL (d = 0.32), velocity ST in ROAD (d = 0.23), and for stride time DT in both groups (d = 0.43 in TRAIL; d = 0.45 in ROAD) were calculated.

### 3.3. Agility

Both groups improved their *t*-test performance by 4.6% (TRAIL) and 6.8% (ROAD), respectively. Yet, no significant change over time or between groups was observed. Effects from the intervention on agility are shown in Table 5.

### 3.4. Strength

Gains in isokinetic concentric leg strength were only recorded in knee extension TW (8.2%) and knee flexion TW (11.8%) for TRAIL, and knee extension TW (1.6%) as well as ankle dorsi flexion TW (1.9%) for ROAD. Thereof, only knee flexion TW in favor of TRAIL resulted in a close to significant between-group difference over time (*p* = 0.06; ηp2 = 0.19; d = 0.25). This finding was reinforced by a 76% likely probability of a substantial worthwhile effect according to the MBI approach. A significant negative time-effect in ankle plantar flexion PT (*p* = 0.02; ηp2 = 0.29) was recorded for ROAD. All other strength measures showed small declines between pre- and post-testing, as shown in Table 6.

### 3.5. VO_2_max

The results of the aerobic endurance testing (VO_2_max pred) show the greatest probability for a substantial beneficial effect between pre- and post-testing with 97% in favor of TRAIL. These findings are supported by the calculated Cohen’s d effect sizes (d = 0.95 in TRAIL; d = 0.53 in ROAD). Time-effect (*p* = 0.14) and between-group differences (*p* = 0.13) did not reach statistical significance. Results for VO_2_max pred are depicted in Table 7.

## 4. Discussion

This is the first study that comparatively investigated the impact of trail running versus road running on neuromuscular performance parameters in healthy adults. We hypothesized that running on natural trails would lead to more pronounced improvements in static and dynamic balance, gait patterns, agility, and leg strength between pre- and post-testing compared to road running. This assumption was based on previous findings which have shown that the navigation of the body on varying surface densities, inclines and speeds evoked higher muscle activation and coordination as opposed to moving on more firm and flat terrain [23,24,25,26,45,46]. Greater physiological strain on softer terrain is associated with a greater degree of energy absorption by the training surface that results in a loss of elastic energy, followed by greater concentric work and overload stimulus in the lower-limb muscles [26,45]. Against this background, we expected gains in concentric quadriceps and hamstring muscle strength as well as in ankle strength and stability in favor of TRAIL from navigating in uneven terrain. However, according to the BIODEX isokinetic concentric leg strength testing, knee flexion TW was the only parameter that resulted in close to significant improvements. On the other hand, for ankle dorsi flexion PT, a significant negative time-effect was recorded. A possible explanation for this decrease could be found in a reduction in ankle work and range of motion that has been seen when running on uneven and unpredictable terrain in order to stabilize the joint [26]. The fact that all other strength measures showed small declines between pre- and post-testing might be attributed to fatigue as a result of the newly increased exercise routine. It is also possible that the reduced strength outcomes especially for PT values are a consequence of endurance training-specific adaptations. When interpreting leg strength results, baseline differences and high standard deviations in both groups should be taken into account. Especially in TRAIL, large discrepancies in strength scores among subjects in pre- and post-testing were observed. Another factor that added to these inconsistencies is the fact that most participants from both groups had no experience in resistance training, much less with the applied strength-testing device. The lack of experience might have influenced the test performances.

We found no statistically significant differences in the rANOVA analysis between TRAIL and ROAD for static and dynamic balance measures. But a significant time-effect between pre-and posttest was calculated (*p* = 0.001, ηp2 = 0.46) for the BESS test. In addition, large (d = 1.2) and moderate (d = 0.5) effect sizes for Cohen’s d for TRAIL and ROAD respectively indicate potential balance improvements from running, especially on trails. In a review on sports participation and balance performance, Hrysomallis et al. [47] stated that athletes generally have a superior balance ability compared to control subjects as a result of repetitive experience and improved motor responses to proprioceptive and visual cues. Additionally, the same authors observed improved coordination, strength and range of motion. However, it remains unclear whether proprioception can actually be improved by exercise or if athletes just become more skilled at reacting to sensory cues. In a study on functional fitness gains through various types of exercise in older adults, Takeshima et al. [48] reported improvements in dynamic balance (functional reach test) in all intervention groups (balance, aerobic, and resistance training). They also predicted that training on unstable surfaces not only leads to improvements in balance but also in lower-body strength due to greater muscle activation when counteracting increased sway following unexpected perturbations. A few other studies report improvements in locomotion in older adults after aerobic training interventions involving walking, treadmill walking, jogging, and step aerobics [19]. The results of the BESS test in this pilot study support previous findings that physical exercise, specifically running, may have a positive influence on balance. Nevertheless, benefits from running for dynamic and functional balance could not be proven with the administered tests for the lack of significant results in the Y-Balance test and gait analysis.

Despite the close relationship between balance and gait performance in regards to fall- and injury-risk factors [14,19,26,47,48,49], the spatiotemporal gait analysis in this study showed no notable characteristics or changes in any parameter for either TRAIL or ROAD. rANOVA, Cohen’s d, as well as MBI calculations show inconsistent results and no conclusions can be drawn about the influence of trail or road running on gait stability. Likewise, no statistically significant differences for time or between groups were recorded in agility performances. Nevertheless, most participants achieved faster T-test times after the intervention and demonstrated noticeably increased confidence and security levels in their sprint performances. Increased confidence levels and sprint ability might result in an overall increased gait stability and thus reduce fall risk. When discussing the lack of evidence for gait and agility in this study, testing devices and procedures need to be considered. More task-specific trials might elicit more pronounced changes.

Aerobic endurance testing showed the highest probability for a substantial worthwhile effect in favor of TRAIL (97%, very likely) together with a large Cohen’s d effect size (d = 0.95). Relative VO_2_max outcomes from the gas analysis test improved by 23.1% and 13.7% from pre- to post-testing for TRAIL and ROAD, respectively. Still, time-effect (*p* = 0.14; ηp2 = 0.12) and between-group differences (*p* = 0.13; ηp2 = 0.13) did not reach statistical significance. Moreover, big baseline differences (24.3% higher in ROAD) need to be considered when interpreting predicted maximal oxygen consumption. Lower baseline values in TRAIL might have facilitated the larger responses to the training intervention in that group. Even so, it is probable that trail running may elicit greater benefits for cardiovascular fitness. Several studies [26,50,51,52,53] documented that running on natural surfaces such as irregular trails required a higher energy expenditure and metabolic cost, which translated to a higher training intensity and higher aerobic training adaptations. However, recorded RPE from the running logs revealed no group differences (4.6 ± 1.1 for TRAIL; 4.9 ± 0.8 for ROAD), an interesting finding if greater energy expenditure is realized on TRAIL versus ROAD without a concurrent rise in RPE. Therefore, TRAIL could be a strategy or modality for advanced energy output and weight loss, leading to better motor control at a lower perceived exertion.

To date, a lot of research regarding neuromuscular adaptations from running has focused on different types of footwear or foot strike patterns and related kinematic, metabolic, and biomechanical parameters of the lower limb, as well as running-related injuries [20,24,54]. Various research groups examined the effects of training on different outdoor terrain, mainly focused on grass or sand surface [23,45,55,56,57], or defined trail running as an ultra-endurance activity. In this understanding, Easthope et al. [58] analyzed performance levels between young and older master runners in a 55-km ultra-endurance trail run. They observed equal performances in both groups despite structural and functional age-related alterations and confirmed that the decline in physical performance can be prevented with regular endurance training such as running. In a study that compared the different effects of concrete road, synthetic track, and woodchip trail on dynamic stability and loading in runners, Schütte et al. [22] revealed significant performance differences from a biomechanical perspective. Running on woodchip trail altered measures of dynamic stability and lower-limb musculature compared to running on concrete road due to compression and displacement of the woodchips under the foot causing destabilization and directional shift with each stride. Similarly, Boey et al. [59] looked at running on concrete, synthetic running track, grass, and woodchip trail at two different speeds and the different vertical impacts on the lower leg. Their results showed that running on woodchip trails and at a slower speed, reduced the injury risk at the tibia.

Running related injuries (RRI) of the lower extremities are a common negative side effect in runners [60,61]. The prevalence is usually higher for overuse musculoskeletal injuries than for acute injuries [21,60,62]. There is a large heterogeneity of injuries that originates from different methods and definitions when evaluating RRI [21,60]. Among the most commonly reported RRIs in the literature are to the Achilles tendon, plantar fascia, calf muscle, knee, meniscus, shin, foot, ankle, hip/pelvis, lower back, hamstring, and thigh [21,60,63,64]. Risk factors for RRI appear to be previous injuries to the same anatomical area, high training loads and little running experience [64,65].

In the current study, 5 out of 33 people reported an injury during the 8-week intervention that prevented them from completing the training program. Affected body sites and type of injuries are all in line with the formerly reported common injury types and risk factors. Two participants from TRAIL developed reoccurring overuse injuries (i.e., knee and lower back) that had probably not been fully and appropriately cured. The other participants suffered from tibial stress syndrome (1 in TRAIL) and ankle sprains (1 in TRAIL; 1 in ROAD). The recorded amount and type of injuries in this study seem to reinforce the fact that previous injuries, little running experience, and an increase in training load within a relatively short amount of time may be risk factors for RRI. Meanwhile, as stated by Taunton et al. [64], previous activity, cross-training and running surface appear to be non-significant injury risk factors for either gender. Interestingly, 4 of the 5 injured subjects in this study were part of the trail running group, which contrarily seems to imply a connection between surface and injury prevalence. Trail running might be more strenuous for physiological parameters due to its specific surface characteristics and the resulting challenges for involved muscle groups and the metabolic system. Therefore, running on natural and more compliant trails may be more likely to cause overuse injuries in an untrained population. Despite the mentioned risk factors, authors agree that health benefits from running outweigh the related risks and costs of RRI [21].

Limitations to this study are the small group sizes and baseline differences between groups in VO_2_max pred and certain strength parameters, as well as the fact that the running intervention itself was not supervised and subjects performed most of the training units individually. Consequently, even though participants were instructed to exercise at a comfortable, moderate to somewhat hard intensity (3–4 on the Borg CR-10 scale), it is possible that some trained at intensities that were too high for their level of fitness. Additionally, the program was based on running time and not distance, which may have resulted in a different training volume dependent on different training pace among individuals. The training log was a way of controlling for these interferences. Regarding adherence, a slightly lower attendance rate in the trail running group was expected since trails require more effort and planning to access and may become impassable in bad weather or darkness. As a final point, MBI’s should be interpreted carefully, especially implications drawn from them, and one should be mindful of how the performed tests may have related to the intervention.

## 5. Conclusions

The results of this training intervention show no statistically significant between-group differences. This suggests that benefits derived from running on uneven and soft natural terrain as opposed to a more flat and concrete road surface in respect to static and dynamic balance, gait, agility, and lower limb strength should not be overrated. Based on current knowledge and the outcomes of this study, no well-founded recommendations for an integrative training approach in regard to trail running and the prevention of falls and fall-related injuries can be given. More research is needed on the influence of running on trails or similar natural surfaces on different neuromuscular performance parameters.

Nevertheless, the findings of this intervention indicate slightly more beneficial tendencies for balance and leg strength improvements when running on trails as opposed to road; and, therefore, potential benefits for the prevention of falls and fall-related injuries. While a significant time-effect between pre- and post-testing in static balance was recorded for both groups (*p* = 0.001, ηp2 = 0.46), the trail running group also showed large effect sizes (d = 1.2) for static balance, compared to only moderate effect sizes (d = 0.5) in the road running group. Trail running also seems to have positive impacts on upper leg strength performance, which is indicated by gains in knee extension (8.2%) and flexion (11.8%) total work and a close to significant between-group difference over time (*p* = 0.06; ηp2 = 0.19; d = 0.25) in knee flexion TW.

For more detailed and specific results, future studies should target larger group sizes of recreational runners within smaller age ranges and in a longitudinal approach over a longer time period. Moreover, the scope of the intervention should be limited to one particular neuromuscular parameter. Thereby, the combined effects for cardiovascular and neuromuscular performance factors from running on different surfaces might be disentangled more clearly. Finally, repeating this study in an older, untrained population and tracking at-home falls throughout a pre-determined follow-up period (e.g., over 5-years) post intervention could yield more precise commentary regarding TRAIL’s effectiveness in or lack of promoting better neuromuscular coordination.

## Figures and Tables

**Table 1 ijerph-20-04501-t001:** Demographic data at baseline.

	TRAIL ^1^ (n = 10, 6 fem)	ROAD ^1^ (n = 10, 7 fem)	Total (n = 20)
Female/male (n)	6/4	7/3	13/7
Age (years)	33.2 ± 6.8	29 ± 10.5	31.3 ± 8.8
Height (cm)	171.1 ± 8.0	170.9 ± 6.6	171 ± 7.3
Weight (kg)	77.4 ± 17.6	74.5 ± 15.6	76.1 ± 16.5
BMI (kg/m^2^)	26.2 ± 4.1	25.4 ± 4.5	25.8 ± 4.3
Physical Activity (min/week)	1904.8 ± 957.5	2105.3 ± 1679.5	2000.5 ± 1445.6

^1^ Values are mean (±SD). TRAIL = trail running group. ROAD = road running group.

**Table 2 ijerph-20-04501-t002:** Training load of trail and road physical activity interventions. Values are mean ± SD.

Training Load	TRAIL (n = 10, 6 fem)	ROAD (n = 10, 7 fem)	Total (n = 20, 13 fem)
Weeks (n)	9.0 (0.7)	9.6 (0.8)	9.3 (0.8)
Trainings (n)	26.5 (1.7)	27.9 (2.6)	27.2 (2.3)
Sessions/week (n)	3.0 (0.3)	2.9 (0.4)	2.9 (0.3)
Time/session (min)	35.4 (1.7)	34 (1.6)	34.7 (1.8)
Intensity/session (RPE)	4.9 (1.1)	4.6 (0.8)	4.8 (1)
Total Training Time (min)	938.4 (68.3)	946.7 (82.9)	942.5 (74.8)

**Table 3 ijerph-20-04501-t003:** Effect on balance of an 8-week trail and road running training intervention.

	TRAIL			ROAD			rANOVA			
TEST	Pre-Test	Post-Test	Cohen’s D	Pre-Test	Post-Test	Cohen’s d	Time	ηp2	Group × Time	ηp2
BESS	12.4 (2.7)	9.5 (2.3)	1.2	11.6 (3.9)	10.0 (2.8)	0.5	*p* = 0.001	0.46	*p* = 0.38	0.05
YBT left	94 (8.2)	95.9 (7.7)	0.25	94.4 (7.3)	94.7 (7.2)	0.04	*p* = 0.19	0.10	*p* = 0.31	0.06
YBT right	93.8 (8.5)	96.4 (8.2)	0.31	92.4 (8.1)	93.8 (7.4)	0.18	*p* = 0.15	0.12	*p* = 0.3	0.06

Values are mean (±SD); statistical significance level is set at *p* < 0.05.

**Table 4 ijerph-20-04501-t004:** Effect on spatio-temporal gait characteristics.

		TRAIL			ROAD			rANOVA			
		Pre-Test	Post-Test	Cohen’s d	Pre-Test	Post-Test	Cohen’s d	Time	ηp2	Group × Time	ηp2
Stride time [s]	ST	1.1 (0.1)	1.1 (0.1)	0.01	1.1 (0.1)	1.1 (0.1)	0.52	*p* = 0.7	0.009	*p* = 0.37	0.05
	DT	1.2 (0.1)	1.2 (0.1)	0.43	1.2 (0.1)	1.2 (0.1)	0.45	*p* = 0.89	0.001	*p* = 0.35	0.05
Stride length [m]	ST	1.4 (0.1)	1.4 (0.1)	−0.09	1.4 (0.1)	1.4 (0.1)	0.09	*p* = 0.65	0.01	*p* = 0.35	0.05
	DT	1.3 (0.1)	1.3 (0.1)	0.19	1.3 (0.1)	1.3 (0.1)	−0.17	*p* = 0.84	0.002	*p* = 0.37	0.05
Velocity [m/s]	ST	1.3 (0.2)	1.3 (0.2)	−0.06	1.3 (0.2)	1.4 (0.2)	0.23	*p* = 0.8	0.006	*p* = 0.34	0.05
	DT	1.1 (0.2)	1.2 (0.2)	0.32	1.2 (0.2)	1.2 (0.2)	0.06	*p* = 0.45	0.03	*p* = 0.3	0.06

Values are mean (±SD); ST, single task condition; DT, dual task condition; statistical significance level is set at *p* < 0.05.

**Table 5 ijerph-20-04501-t005:** Effect on agility.

	TRAIL			ROAD			rANOVA			
Agility	Pre-test	Post-test	Cohen’s d	Pre-test	Post-test	Cohen’s d	time	ηp2	group × time	ηp2
*t*-test [s]	15.6 (3.2)	14.9 (2.4)	0.26	15.1 (3.2)	14.1 (2.4)	0.36	*p* = 0	0.69	*p* = 0.15	0.12

Values are mean (±SD); significance level is set at *p* < 0.05.

**Table 6 ijerph-20-04501-t006:** Effect on strength.

	TRAIL			ROAD			rANOVA			
	Pre-Test	Post-Test	Cohen’s d	Pre-Test	Post-Test	Cohen’s d	Time	ηp2	Group × Time	ηp2
**Knee**										
KE PT (Nm)	175.5 (74.6)	172.7 (68)	−0.04	163.9 (55.5)	154.3 (49.1)	−0.18	*p* = 0.06	0.2	*p* = 0.37	0.05
KF PT (Nm)	96.3 (41)	91.6 (37.6)	−0.12	80.3 (25.7)	77.0 (21.9)	−0.14	*p* = 0.06	0.2	*p* = 0.77	0.005
KE TW (J)	870.9 (419.4)	945.3 (406.3)	0.18	833.9 (283.8)	847.7 (303.6)	0.05	*p* = 0.56	0.02	*p* = 0.21	0.09
KF TW (J)	487.2 (245.4)	548.3 (244.8)	0.25	447.8 (151.3)	446 (164.5)	−0.01	*p* = 0.67	0.01	*p* = 0.06	0.19
**Ankle**										
PF PT (Nm)	60.2 (32.3)	55.8 (26.5)	−0.15	50.8 (18.1)	45 (16.5)	−0.33	*p* = 0.02	0.29	*p* = 0.49	0.03
DF PT (Nm)	25.5 (7.4)	24.1 (7.7)	−0.18	24.2 (6)	23.7 (5.8)	−0.09	*p* = 0.5	0.03	*p* = 0.5	0.03
PF TW (J)	167 (93.1)	166.6 (86.5)	−0.004	140.6 (54.1)	134.7 (55.8)	−0.106	*p* = 0.14	0.12	*p* = 0.57	0.02
DF TW (J)	104.6 (28.3)	97.6 (31.6)	−0.24	94.8 (20.3)	96.6 (26.4)	0.07	*p* = 0.9	0.001	*p* = 0.28	0.07

Values are mean (±SD); KE, knee extension; KF, knee flexion; PF, plantar flexion; DF, dorsi flexion; PT, peak torque; TW, total work; i.f.o., in favor of; statistical significance level is set at *p* < 0.05.

**Table 7 ijerph-20-04501-t007:** Effect on VO_2_max of an 8-week trail and road running training intervention.

	TRAIL			ROAD			rANOVA			
	Pre-test	Post-test	Cohen’s d	Pre-test	Post-test	Cohen’s d	time	ηp2	group × time	ηp2
pred. VO_2_max	28.4 (6)	35.8 (9.2)	0.95	35.3 (8.8)	40.5 (10.4)	0.53	*p* = 0.14	0.12	*p* = 0.13	0.13

Values are mean (±SD); i.f.o., in favor of; statistical significance level is set at *p* < 0.05.

## Data Availability

Data for this project are maintained by the primary authors—S.N.D. and L.D.—on personal and password protected laptops.

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
