# Peer review of "Effects of Trail Running versus Road Running—Effects on Neuromuscular and Endurance Performance—A Two Arm Randomized Controlled Study"

_ijerph, 2023, doi:10.3390/ijerph20054501_

Round 1

Reviewer 1 Report

Effects of Trail Running Versus Road Running – Effects on 2 Neuromuscular and Endurance Performance – a Two Arm Ran-3 domized Controlled Study

The main limitation of that study is the sample size and their profile. Why did the authors decided to analyse sedentary participants and not experienced or recreational runners with some experience in trail and road running?

And, why did author decided to divide the group and not to analyse the whole group (n= 20) in the two conditions?

Line 22: stride time single task (d=0.52) that term doesn’t appears before in the test. What are the authors referring to?

What did the authors refereed with “Possible effects” (line 22)?

Why did the authors interpret “the results suggested slightly more beneficial tendencies in favor of TRAIL” line 24?

A nivel general, falta un nexo de unión entre los argumentos propuestos en la introducción y las pruebas y resultados obtenidos. Faltaría justificar mejor la inclusión de los test utilizados.

Introduction:

No need so many references for that kind of affirmations.

Line: 33

If it is not the object of study, authors are recommended to avoid speculation and not include this type of statements/phrases in the text, as they divert attention and generate incorrect expectations. Lines 57-60.

These lines correspond to the methodology section, they should not be in the introduction section: lines 64-70.

Avoid these kinds of speculative claims. Lines: 73-74.There is not objective describe at the end of the introduction section.

There is not link between the methodology and the test employed in the study. There is a lack of connection between the objectives (not presented in the test) and the rationale of the introduction.

Meterial and Methods

Line 84: please remove webpage and add a reference.

Lines 85-86: That information is not needed. Additionally, clarify why that inclusion criteria and add some reference for that. Why sedentary population?

Line 91-92: add references to the physical questionnaires and name them in that section of the manuscript.

Please, add the training program characteristics as a supplementary file. More information about the training load control and the differences between programs are required. More information about time expended in training zones, unevenness, intensities, control strategies, distances, etc.

Table 1 it’s not required, as that doesn’t represent the final sample of the study. Please, change it by the data from the final sample. Participant number doesn’t match within the text, please, adapt the text accordingly.

Table 1: Change Walking by Physical Activity or similar.

No comment has been made about the fact of inclusion both, men and women, as a one simple sample. There were differences between men and women? What was the final proportion of the n=10 groups?

Line 94-95: Change World Medical 94 Association, 2013 by a reference.

Lines 110-112: please remove all the laboratory information, as that is not relevant for the study.

Lines 116-121: Should be moved to the participant’s section to clarify the final sample size.

Did the author employed any software to calculate sample size? Add the methods that authors employed to calculate the sample size and the power estimation.

Line 122:123: I can’t see the study flow. Please add to the manuscript and delate the web page.

Section 2.3. Physical activity questionnaires: Please add references to each questionnaire.

Why did you decided to include participants with two risk factors related to cardiovascular diseases? I other studies, that kind of participant were drop off the study. How many participants needed to go to medical exam to participate in the study?

Lines 143-147 are not heart rate and blood pressure. What tool did you employed for heart rate, blood pressure, body height and weight? Please add that information in the text.

Why did you register the leg length by this way? And for what? Other studies made that measurement from the iliac crest to the medial malleolus.

Static balance testing: how it was distributed in time the 3 conditions and the 2 surfaces?

In the Y balance test, why did you only performed one practice test? It should be an knowledge effect.

No randomization of the order of the direction was followed. Why? Why did you extract only the mean reach, and not each component (plus the mean)?

Line 201-208, 227: Remove all the references to yards and mi units.

Line 234: add a reference.

Statistical analysis: Did the authors performed any test to check the distribution? Shapiro-Wilk or kolmogorov-smirnov test?

RESULTS:

Lines 265-270: This information is repeated.

Lines 271-274: please add p value and effect size.

Lines 276-278: interpretations should be moved to the discussion section.

Agility test: Please add the exact p value to the main differences. The SD for both groups make complicate to understand the differences described.

Strength:

-        Only report the significant results, not the close to significance.

-        There was a reduction in strength, what could be the reason?

-        How the differences between groups at the baseline test could affect the results?

-        The table 6 is complex to read due to the number of abbreviations and line breaks. Reduce the data (effect sizes and p values) to make the data easy to read. Include differences by using signs in the table itself (superscripts).

CONCLUSIONS:

The authors introduce certain speculations that are not refuted by the results of the study, which is why the authors should eliminate any type of forced interpretation and/or interpretation not derived directly from their results.

TABLES:

Table are complex to read due to the number of abbreviations and line breaks. Reduce the data (effect sizes and p values) to make the data easy to read. Include differences by using signs in the table itself (superscripts).

Author Response

Please see the attached file with our responses. We appreciate the time and effort you put forth in your initial review of our work. 

Reviewer 2 Report

Dear authors,

I feel the work is well done. I have some minor comments.

p. 2 lines 88-89: how did you choose the cutoff at 2 times per week? Were there limitations regarding the type of activity?

Static balance testing: how were angles measured or was it a rough estimation?

p. 5 line 222: were 60s enough in between maximal trials?

p. 5 lines 233-234: please add references.

Discussion: please speculate about the influence of running experience while still considering the recreational level. Do you think that if participants had some running experience the outcomes would have been different?

Best regards

Author Response

Please see the attached document with our responses to your review. Thank you for your time and efforts!
